# Overcoming Multi-step Complexity in Multimodal Theory-of-Mind Reasoning: A Scalable Bayesian Planner

Chunhui Zhang [1]   Zhongyu Ouyang [1]   Kwonjoon Lee [2]   Nakul Agarwal [2]
Sean Dae Houlihan [1]   Soroush Vosoughi [1 †]   Shao-Yuan Lo [2 †]

## Abstract

Theory-of-Mind (ToM) enables humans to infer mental states—such as beliefs, desires, and intentions—forming the foundation of social cognition. Existing computational ToM methods rely on structured workflows with ToM-specific priors or deep model fine-tuning but struggle with scalability in multimodal environments. They remain trapped within the gravitational pull of multi-step planning complexity, failing to generalize as task demands increase. To overcome these limitations, we propose a scalable Bayesian ToM planner. It breaks down ToM complexity into stepwise Bayesian updates. Meanwhile, weak-to-strong control specializes smaller LMs to refine ToM-specific likelihood estimation, transferring their ToM reasoning behavior to larger LMs (7B to 405B) for social and world knowledge integration. This synergistic approach enables scalability, aligning large-model inference human mental states with Bayesian principles. Extensive experiments demonstrate a 4.6% improvement in accuracy over state-of-the-art methods on multimodal ToM benchmarks, including unseen scenarios, establishing a new standard for modeling human mental states in complex environments.

## 1. Introduction

Theory-of-Mind (ToM) is a cornerstone of human social cognition, enabling individuals to attribute and infer mental states in themselves and others. This capacity underpins the recognition that others may have perspectives dis-

See the contributions of student and senior authors. [1]Dartmouth College, Hanover, NH, USA [2]Honda Research Institute USA, San Jose, CA, USA. Correspondence to: Soroush Vosoughi <soroush.vosoughi@dartmouth.edu>, Shao-Yuan Lo <shao-yuan_lo@honda-ri.com>.

*Proceedings of the 42nd International Conference on Machine Learning*, Vancouver, Canada. PMLR 267, 2025. Copyright 2025 by the author(s).

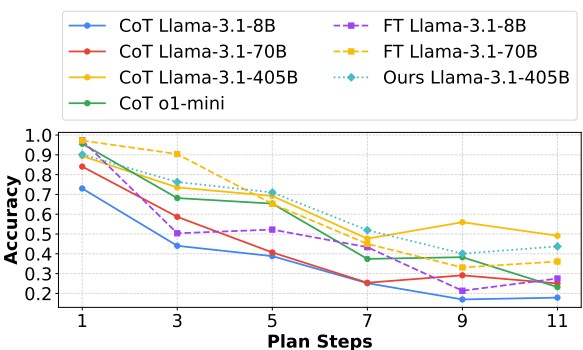

Figure 1. Comparison of models on planning tasks in VirtualHome (a simulator used in multimodal ToM). As planning steps increase, smaller models (e.g., Llama3.1-8B, 70B) and inference-time scaling (e.g., o1-mini, CoT) fail to sustain accuracy. Only larger models (e.g., Llama3.1-405B) maintain performance, demonstrating that sustaining accuracy requires model scaling.

tinct from one's own (Dennett, 1988; Gopnik & Wellman, 2012). Similarly, ToM tasks in AI involve perceiving observable cues—such as an agent's actions and surrounding context—to predict their goals and beliefs. Equipping AI systems with ToM capabilities could unlock their potential for human-like social understanding and interactions (Lake et al., 2017; Wu et al., 2021; Ma et al., 2023).

Existing approaches to ToM reasoning follow two main strategies: *(i)* structured planning workflows with ToM-specific priors (Baker et al., 2017; Jara-Ettinger, 2019; Shu et al., 2021), and *(ii)* integrating these priors into language models (LMs) via specialized training (Rabinowitz et al., 2018; Shu et al., 2021; Sclar et al., 2022; Jin et al., 2024). The challenges extends beyond method designs—ToM environments demand multimodal reasoning, integrating visual, textual, and contextual inputs into coherent mental state inferences. Understanding and predicting the goal of agent planning behaviors require models to reason across multiple steps while integrating multimodal cues.

However, these challenges introduce generalization barriers, making it unclear whether standard approaches can

scale effectively. To examine these challenges, Fig.1 compares the performance of various models in planning tasks with the physical simulator used by multimodal ToM benchmarks. While inference-time scaling methods—such as CoT reasoning, o1 systems, and fine-tuning experts—provide incremental improvements, they fail to maintain accuracy as task complexity increases. Only larger models, such as Llama3.1 405B, sustain performance over increasing planning steps, while smaller LMs exhibit rapid accuracy degradation. These scalability challenges stem from two fundamental factors: *First*, a **reasoning boundary** exists, limiting the number of **effective reasoning steps** regardless of the method used. Prior studies (Chen et al., 2024; Zhang et al., 2024; Ye et al., 2025; Gao et al., 2025) show that CoT reasoning and o1/r1-like scaling methods plateau in effectiveness as task complexity grows, leading to diminishing returns. *Second*, multimodal ToM reasoning **relies heavily on integrating broader social and world knowledge**, which correlates strongly with model scale. Unlike closed-form logic tasks, ToM reasoning in complex environments requires grounding in extensive, diverse contexts. As shown in Zhang et al. (2024); Yu et al. (2024); Sun et al. (2024); Gao et al. (2024); Yuan et al. (2025); Diao et al. (2025b); Zhang et al. (2025b), smaller LMs struggle to generalize effectively due to their limited pretraining capacity to encode and utilize world knowledge. These findings emphasize that simple fine-tuning or inference-time scaling alone is insufficient to improve ToM reasoning at scale. Instead, sustained performance requires both structured frameworks and models with expansive generalization capabilities, as demonstrated by Fig.1, deviating from the existing methods.

Motivated by these scalability challenges, we introduce a more *scalable* Bayesian ToM solution that directly addresses inference-time reasoning complexity through Bayesian Inverse Planning (BIP) (Baker et al., 2007; 2009; 2017; Shum et al., 2019; Jin et al., 2024) and overcomes world knowledge dependency by scaling LMs up to **405B** for ToM-specific generalization. *First*, BIP mitigates the **reasoning boundary** by decomposing multimodal ToM reasoning into **modular, stepwise Bayesian updates**—such as state transitions, belief updates, and action likelihoods. This structured framework refines beliefs and hypotheses iteratively, ensuring tractability even in complex environments. *Second*, overcoming the reliance on broad social and world knowledge requires **scaling to larger LMs**, as illustrated in Fig.1. Prior approaches (Jin et al., 2024) relied on smaller LMs for likelihood estimation, but their limited capacity hindered generalization in rich ToM settings. Our weak-to-strong control mechanism enables smaller, post-trained LMs to specialize in ToM-specific tasks and transfer these learned behaviors to larger LMs (up to **405B**) during inference. This innovation allows the *larger LM to serve as the primary policy model*, leveraging its extensive world knowledge

while maintaining Bayesian consistency for stable and interpretable reasoning. The *theoretical foundation* of our approach is established by Theorem 1, which formalizes its effectiveness through *KL divergence analysis*. Empirical results demonstrate that our scalable solution enhances the generalizability of Bayesian inference and achieves a 4.6% accuracy improvement over state-of-the-art methods on multimodal ToM tasks, including unseen scenarios, setting a new standard for modeling ToM in complex environments.

## 2. Preliminaries

**Behaviour modelling: a Markov decision process formulation** The behaviour of an agent can be formulated as a forward generative model based on a Partially Observable Markov Decision Process (POMDP), defined by the tuple $\langle S, A, \mathcal{T}, G, R, \Omega, O, \gamma \rangle$ (Kaelbling et al., 1998; Jin et al., 2024). Here, $s^t \in S$ and $a^t \in A$ represent the state and action at time $t$, respectively. $\mathcal{T}(s^t|s, a)$ denotes the state transition probabilities. The goal $g \in G$ determines the reward $r^t = R(s^t, a^t, g)$. The agent's observation $o^t \in \Omega$ is obtained via the observation function $o^t = O(s^t)$. The discount factor is $\gamma \in (0, 1]$. Crucially, the agent's belief, $b(s)$, is a probability distribution over the state. This belief is dynamically updated during belief evolution $P(b^\tau \mid b^{\tau-1}, s^\tau)$, where $b(s)$ is factorized into probabilities over the possible locations of individual objects.

In our practice, $b^0$ is initialized as all possible object locations at the start. At each step $\tau$, if an object is observed inside a container, $b^\tau$ is updated to include the container; otherwise, unobserved locations are removed from $b^\tau$. $P(b^\tau|b^{\tau-1}, s^\tau)$ can be approximated using likelihoods from an LM.

**Inverse inference: from observed behaviours to mental states** Bayesian inverse planning (BIP) infers an agent's goals and beliefs by **inverting** the forward POMDP model (Baker et al., 2017). Given observed states $s^{1:t}$ and actions $a^{1:t-1}$, the posterior probability of a goal $g$ and belief $b^t$ is expressed as:

$$P(g, b^t|s^{1:t}, a^{1:t-1}) \propto \prod_{\tau=1}^{t} \pi(a^\tau|g, b^\tau)$$
$$P(b^\tau|b^{\tau-1}, s^\tau)P(b^0)P(g), \quad (1)$$

where $\pi(a^\tau|g, b^\tau)$ represents the agent's policy—the probability of taking action $a^\tau$ given its goal $g$ and belief $b^\tau$. This process combines the likelihood of actions and belief updates with prior probabilities, refining beliefs incrementally as new evidence is observed. To compare hypotheses about

an agent's goals, we evaluate their relative log-likelihoods:

$$
\log \frac{P(g_1, b_1^t | s^{1:t}, a^{1:t})}{P(g_2, b_2^t | s^{1:t}, a^{1:t})} = \underbrace{\sum_{\tau=1}^{t-1} \log \frac{\pi(a^\tau | g_1, \hat{b}^\tau)}{\pi(a^\tau | g_2, \hat{b}^\tau)}}_{\text{Prior steps comparison}}
$$

$$
+ \underbrace{\log \frac{\pi(a^t | g_1, b_1^t)}{\pi(a^t | g_2, b_2^t)} + \log \frac{P(b_1^t | \hat{b}^{t-1}, s^t)}{P(b_2^t | \hat{b}^{t-1}, s^t)}}_{\text{Current step comparison}}. \quad (2)
$$

The first term compares cumulative likelihoods across prior steps, while the second term evaluates how well each hypothesis aligns with the agent's latest action and belief update.

In practice, the hypothesis with the highest accumulated likelihood is selected, and $\pi(a^\tau | g, b^\tau)$ can be approximated using likelihoods generated by a language model.

## 3. Bayesian ToM Planner *at Scale*

Our scaled Bayesian planner infers an agent's mental state based on the unified representations about a scene, a person's actions, and the mental state hypotheses from multimodal inputs, and then post-train an LM to conduct contextual inverse symbolic planning, based on unified symbolic representations (Jin et al., 2024). Then, as shown in Fig.2, the LM used in our scaled Bayesian planner is from 7B up to 405B parameters at test-time compute. This approach harnesses the world knowledge and reasoning capabilities of large LM, avoiding additional post-training.

**Weak-to-strong controlled large policy model** When augmenting likelihood estimation with the guided large LM's broad capabilities, we scale up the LM used only for inference-time computing and avoid the direct post-training on large LM. The true policy $\pi(a^\tau | g, b^\tau)$ is estimated through a LM ($\pi$)-estimated probability $\tilde{\pi}(a^t | s^t, g, \hat{b}^t)$:

$$
\pi(a^\tau | g, b^\tau) = \tilde{\pi}(a^t | s^t, g, \hat{b}^t) + \varepsilon, \quad (3)
$$

where $\varepsilon$ represents the inherent approximation error. When applied to the Bayesian inverse planning (1), the posterior probability is expressed as:

$$
P(g, b^t | s^{1:t}, a^{1:t-1}) \propto \prod_{\tau=1}^{t} \left[ \tilde{\pi}(a^\tau | s^\tau, g, \hat{b}^\tau) + \varepsilon \right] \cdot
$$
$$
P(b^\tau | b^{\tau-1}, s^\tau) \cdot P(b^0) P(g). \quad (4)
$$

It integrates the approximation into Bayesian planner, considering $s_i, b_i, g_i, a_i$ updates over time.

**Post-training stage: ToM optimization** To specialize the LM's behaviors to estimate likelihoods and reduce approximation error $\varepsilon$ in Eq 3, the post-training stage consists of

two phases: In **instruction tuning** phase, a scenario-specific policy $\pi^{\mathcal{E}_0}$ is refined using an action-policy experience pool $\mathcal{D} = \{(s_i, b_i, g_i, a_i)\}_{i=1}^{N}$, where $s$, $b$, $g$, and $a$ represent states, beliefs, goals, and actions. The tuning objective maximizes the likelihood of observed actions:

$$
\mathcal{L}_{\text{IT}}(\pi^{\mathcal{E}_0}) = -\sum_{i=1}^{N} \log \pi^{\mathcal{E}_0}(a_i | s_i, b_i, g_i). \quad (5)
$$

This step builds a mapping from multimodal environments to goal-directed actions. Then in **preference optimization** phase, LM is further aligned by distinguishing between effective ($a^+$) and ineffective ($a^-$) actions. $a^+$ corresponds to concise, successful outputs, while $a^-$ represents long descriptions or failed actions. The preference loss, modified from DPO (Rafailov et al., 2023), is defined as:

$$
\mathcal{L}_{\text{PO}} = -\mathbb{E}_{(\mathbf{x}, a^+, a^-) \sim \mathcal{D}} \left[ \log \sigma \left( \beta \cdot \Delta \log \pi^{\mathcal{E}} \right) \right]
$$
$$
+ \lambda \cdot \mathbb{E}_{\mathbf{x}, a \sim \pi^{\mathcal{E}_0}} \left[ \log \frac{\pi^{\mathcal{E}_0}(a | s, b, g)}{\pi^{\mathcal{E}}(a | s, b, g)} \right], \quad (6)
$$

where $\Delta \log \pi^{\mathcal{E}} = \log \pi^{\mathcal{E}}(a^+ | s, b, g) - \log \pi^{\mathcal{E}}(a^- | s, b, g)$ represents the log-odds. Here, $\beta$ controls the sharpness of preference learning, while $\lambda$ regularizes deviations from the initial policy $\pi^{\mathcal{E}_0}$, ensuring stable training.

*Inference stage: large policy model with behavioral guidance.* During inference, we leverage the behaviour acquired by the ToM preference-optimized smaller LM to guide the reasoning of a larger, more capable LM. This approach dynamically adjusts the output of the large LM based on the shift observed between a post-trained small LM $\pi^{\mathcal{E}}$ and a naive small LM $\pi^{\mathcal{N}}$. At each inference step $t$, the policy distribution for the redirected large LM is given by:

$$
\bar{\pi}(a^t | s^t, g, \hat{b}^t) = \frac{1}{\bar{Z}} \pi^{\mathcal{L}}(a^t | s^t, g, \hat{b}^t) \frac{\pi^{\mathcal{E}}(a^t | s^t, g, \hat{b}^t)}{\pi^{\mathcal{N}}(a^t | s^t, g, \hat{b}^t)}, \quad (7)
$$

where $\pi^{\mathcal{L}}(a^t | s^t, g, \hat{b}^t)$ represents the policy distribution from the naive large LM. The post-training effect to policy function is *approximated* through the ratio $\frac{\pi^{\mathcal{E}}(a^t | s^t, g, \hat{b}^t)}{\pi^{\mathcal{N}}(a^t | s^t, g, \hat{b}^t)}$, offering an on-the-fly redirecting mechanism. The normalization factor is calculated by $\bar{Z} = \sum_{a^t} \pi^{\mathcal{L}}(a^t | s^t, g, \hat{b}^t) \frac{\pi^{\mathcal{E}}(a^t | s^t, g, \hat{b}^t)}{\pi^{\mathcal{N}}(a^t | s^t, g, \hat{b}^t)}$. It ensures the resulting probabilities being aligned, reflecting both the post-training adjustments and the base likelihood from the larger LM. Our overall method facilitates ToM behaviour transfer from the post-trained small LM ($\pi^{\mathcal{E}}$) to the larger LM ($\pi^{\mathcal{L}}$), scaling the large policy model's capabilities in BIP at inference time.

Below Eq 8 shows this behavior control relies on the learned $\Delta s$ to approximate the scaled gradient $-\eta \nabla_s \mathcal{L}_{\text{CE}}(s_{\pi^{\mathcal{L}}}, y)$ with higher-order terms contributing to the residual error.

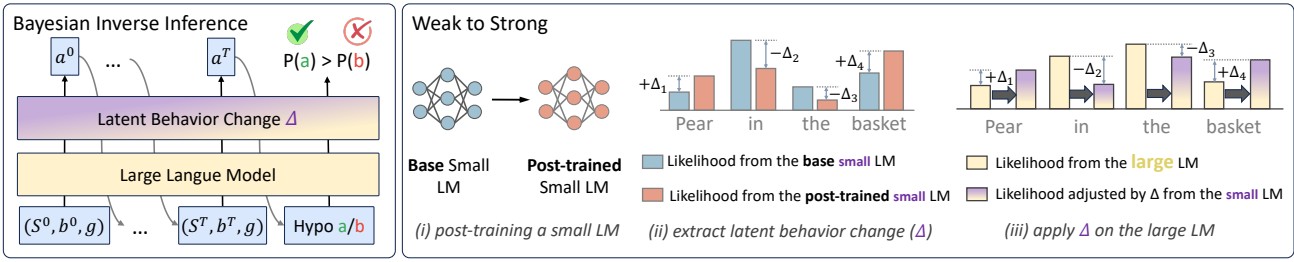

*Figure 2. (left)* The large LM operates as a scaled **policy model** (e.g., 405B) to estimate the likelihood of an agent's actions in dynamic environments, based on multimodal symbolic inputs (video and description). *(right)* The latent reasoning of the large LM is guided by the ToM behaviours from post-trained small LMs, which acts as a weak-to-strong scaling control. Overall, Bayesian inverse planning compares hypotheses about the agent's goal and belief, using the large LM as a policy model to infer ToM.

---

**Theorem 1.** *Let $\pi^*$ be the directly post-trained base model. Suppose the output adjustment $\Delta s(X_t|x_{<t}) = s_{\pi^{\mathcal{E}}}(X_t|x_{<t}) - s_{\pi^{\mathcal{N}}}(X_t|x_{<t})$ approximates the scaled negative gradient of the CE loss for outputs, i.e., $\Delta s \approx -\eta \nabla_s \mathcal{L}_{CE}(s_{\pi^{\mathcal{L}}}, y)$, where $\eta$ is the learning rate. Then, the model $\tilde{\pi}$, defined by $s_{\tilde{\pi}}(X_t|x_{<t}) = s_{\pi^{\mathcal{L}}}(X_t|x_{<t}) + \Delta s(X_t|x_{<t})$, approximates $\pi^*$. The KL divergence between their output distributions is:*

$$D_{KL}(P_{\pi^*} \| P_{\tilde{\pi}}) \leq \frac{\eta^2}{2} \lambda_{\max} \| \nabla_s \mathcal{L}_{CE}(s_{\pi^{\mathcal{L}}}, y) \|_2^2 + \mathcal{O}(\eta^3), \tag{8}$$

*where $\lambda_{\max}$ is the maximum eigenvalue of the Hessian of the cross-entropy loss for the outputs.*

---

$\pi^{\mathcal{E}}$ does not need to strictly approximate the exact loss gradient of $\pi^{\mathcal{L}}$. Instead, $\pi^{\mathcal{L}}$ can use its intrinsic capacity to adapt to ToM scenarios, based only on the approximated $\Delta s$ learned by the small LM. See App C for the proof.

## 4. Experiments

Fig.1 first shows the **scaling benefits** in physical simulated scenarios. To dive into our BIP method in multimodal ToM, for the *strong* component, we scale up the large LMs to 70B and 405B parameters. In contrast, for the *weak* component, we reduce the size of the small LMs from 8B to 4B parameters. First, the results reveal a positive correlation between model size and ToM capabilities, especially when the larger models are guided by the post-trained behaviours of the smaller models. Interestingly, these post-trained behaviours are also effectively captured by smaller LMs. We also illustrate how the large LMs are progressively redirected to the answer space during the Bayesian process.

### 4.1. Setup

**Datasets** *(i) For post-training*, we use MMToM sampled from an apartment environment simulator, Virtual Home (Puig et al., 2018), using the procedural methods described by Jin et al. (2024). The dataset comprises 1,000 procedurally synthesized videos within a realistic household simulator, each annotated with states, goals, beliefs, and actions. *(ii) For evaluation*, we use the MMToM-QA (Jin et al., 2024), an evaluation benchmark aimed at evaluating ToM reasoning over multimodal situations. The dataset consists of 134 videos, each showing a person searching for household objects, with an average of 1,462 frames per video representing approximately 36 human actions. These videos are accompanied by 600 questions (detailed in App. §D.1), evenly divided between the categories of belief inference (with 1.1, 1.2, and 1.3 subtasks) and goal inference (with 2.1, 2.2, 2.3, and 2.4 subtasks). The questions assess the ability of models to infer goals and beliefs jointly.

**Baselines** We include three types of baselines. *For text-only evaluation*, we compare performance in the text-only subset of MMToM-QA using various LMs, including GPT-4 (OpenAI, 2023a), GPT-3.5, Llama2-7B (Touvron et al., 2023), OpenAI-o3-mini, DeepSeek-R1-671B (Guo et al., 2025). Advanced prompting methods, such as SimToM (Wilf et al., 2024) and SymbolicToM (Sclar et al., 2023), which enhance GPT-4's reasoning capabilities, provide additional baselines (e.g., SimToM with GPT-4 and Symbolic-ToM with GPT-4). *For multimodal evaluation*, we include GPT-4V (OpenAI, 2023a), Video-Llama2 (Zhang et al., 2023), and LLaVA (Liu et al., 2023), BIPALM (Jin et al., 2024). *For human*, 180 participants answer 120 randomly sampled questions, covering all question types, as reported by Jin et al. (2024).

**Post-training** We post-train Llama (Touvron et al., 2023; Dubey et al., 2024) as a policy model with LoRA (Hu et al., 2022), as outlined in Tab.7. Following the setup recommended by Jin et al. (2024), we use a learning rate of 1e-3 over 3 epochs. LoRA is configured with a rank of 16 and an alpha value of 32 for the 7B and 8B LMs. For 70B, we use a lower rank of 8 and an alpha of 16.

## 4.2. Main results

Tab.1 uses human performance as the gold standard, with humans achieving a clear lead of 93.0% accuracy on tasks with multimodal input. Among the models, our solution with multimodal input achieves the highest performance, highlighting the critical role of integrating both visual and textual modalities. Comparisons between LMs alone and those augmented with ToM workflows (e.g., SimToM, SymbolicToM, or BIP) further demonstrate the benefits of the ToM workflow. Specifically, the ToM workflow improves performance by **decomposing the complexity** of multimodal ToM reasoning into modular steps. In *belief inference, which is strongly linked to world knowledge*, models like GPT-4 and GPT-3.5 perform exceptionally well, particularly on task 1.1, where GPT-4 achieves an accuracy of 94%. This result underscores the importance of large-scale models in capturing and applying vast amounts of pretrained world knowledge. However, despite their impressive performance in belief inference, these models do not perform as effectively on *goal inference, where adaptation to specific ToM contexts and dynamic environments is crucial*. **This highlights the need for models to be better aligned with the specific requirements of ToM scenarios.** Models with smaller scales, such as those with 6B, 7B, and 13B parameters, face inherent capability limitations, which restrict their performance on belief inference tasks, particularly when compared to larger models like GPT-4 on task 1.1. However, these smaller models, such as BIPALM w/ GPT-J-6B and Llama2-7B, benefit from post-training specifically designed for ToM scenarios. This allows them to perform better on goal inference tasks, where understanding and adapting to scenario-specific environmental dynamics is essential. **Despite their size constraints, these models demonstrate the value of targeted post-training in compensating for the lack of large-scale pretrained knowledge.** Our approach goes beyond seesaw effects in prior methods and has both strengths: while its strong component leverages the extensive world knowledge embedded in large pretrained models, also its weak component incorporates post-training to the ToM contexts and environmental dynamics required. This dual advantage allows a balanced performance across both task types (belief & goal inference), **with an overall 81.3% accuracy on multimodal tasks and exhibits a 4.6% improvement over the existing best baseline.**

## 4.3. Stronger Large LMs enhance likelihood estimation

In the Bayesian framework, we explore the role of LMs in likelihood estimation and examine how their scale and post-training affect performance across various ToM tasks. According to Tab.2, *(i)* **our results demonstrate a positive correlation between LM size and ToM task performance.** For instance, in the zero-shot setting of Llama3.1, the 405B model achieves an accuracy of 69.43%, outperforming both

the 8B model (65.19%) and the 70B model (66.62%). Notably, the performance of the 405B model approaches that of the post-trained Llama2-7B. Furthermore, the improvement from 70B to 405B suggests that the benefits of scaling have not yet reached saturation, indicating potential for further gains with larger models. *(ii)* **Post-training significantly enhances LMs' performance on ToM tasks, even when larger LMs already perform well in zero-shot scenarios.** This effect is consistent across model sizes, from smaller models such as 7B/8B to larger models up to 70B, regardless of the specific version (Llama2, Llama3, or Llama3.1). For belief inference tasks, which are closely tied to world knowledge, post-training helps align the large models' knowledge more precisely with the input questions. For goal inference tasks, which are linked to environmental dynamics, post-training refines the models' atomic-level reasoning (i.e., predicting $a$ based on $s, b, g$), resulting in greater improvements compared to belief inference. This suggests that post-training provides a more substantial benefit for tasks that require dynamic reasoning. *(iii)* **Our weak-to-strong control approach** *approximates* **the benefits of direct post-training in Bayesian inference.** When comparing models such as Llama2, Llama3, and Llama3.1, we find that direct post-training on the 70B model, even with adjusted hyperparameters from the 8B model (e.g., reducing the alpha value from 32 to 16), does not produce results as stable as our method. We attribute this to the difficulty of finding optimal hyperparameters for larger models, which require more extensive tuning. In contrast, our weak-to-strong control, which uses a well-trained smaller LM to guide the larger LMs, allows for more consistent improvements without the need for extensive hyperparameter trials.

## 4.4. Scaling-down small LMs are effective controllers

In the Bayesian planner, prior experiments show that post-trained behaviours from small LMs can effectively guide the pretrained capabilities of larger LMs during test time. To further study the role of post-training to weak-to-strong control, Tab.3 investigates whether post-trained behaviours can be learned effectively with reduced computational resources, while the pretrained capabilities of larger LMs are still available. Specifically, we examine whether *downsized* smaller LMs can effectively capture these post-trained behaviours and guide the pre-trained capabilities of larger LMs without compromising performance. We use 8B LMs as baselines in normal size, and we also downsize them to two 4B variants: Llama3.1-Minitron-4B-Width, which reduces the hidden size of each layer; and Llama3.1-Minitron-4B-Depth, which cuts the model depth (Sreenivas et al., 2024). Despite their smaller size, they maintained comparable accuracy in weak-to-strong control. While the 4B-Width LM underperformed the 4B-Depth LM in zero-shot scenarios, its post-training results surpass the 4B-Depth, especially

*Table 1.* Comparisons between humans and models across task types from 1.1 to 2.4 are provided. The best results for each modal setting are highlighted in **bold**. The second best results in multimodality are underlined. Rows of ours are highlighted in color.

| | method | belief inference | | | | goal inference | | | | | all |
| --- | --- | --- | --- | --- | --- | --- | --- | --- | --- | --- | --- |
| | | 1.1 | 1.2 | 1.3 | avg. | 2.1 | 2.2 | 2.3 | 2.4 | avg. | |
| *text only* | *Human* | 96.0 | 95.8 | 81.3 | 91.0 | 85.8 | 76.7 | 65.0 | 68.3 | 74.0 | 82.5 |
| | GPT-4 | 97.0 | 12.0 | 77.0 | 62.0 | 48.0 | 42.7 | 2.7 | 42.7 | 34.0 | 48.0 |
| | SimToM *w/* GPT-4 | 96.0 | 15.0 | 82.0 | 64.3 | 61.3 | 44.0 | 2.7 | 54.7 | 40.7 | 52.5 |
| | SymbolicToM *w/* GPT-4 | **100** | 61.0 | 74.0 | 78.3 | 73.3 | 66.7 | 0.0 | 50.7 | 47.7 | 63.0 |
| | BIPALM *w/* GPT-J-6B | 88.0 | **69.0** | 88.0 | 81.7 | **77.3** | 68.0 | 30.7 | 70.7 | 61.7 | **71.7** |
| | DeepSeek-R1-671B | 92.0 | 50.2 | 72.4 | 71.5 | 68.3 | 44.0 | 45.0 | 48.2 | 51.4 | 61.5 |
| | OpenAI-o3-mini | 83.1 | 47.6 | 62.3 | 64.3 | 64.0 | 38.6 | 38.7 | 46.0 | 46.8 | 55.6 |
| | **Ours (*w/* Llama3.1-405B)** | 90.1 | 70.5 | 87.4 | 82.7 | 68.8 | 75.5 | 75.3 | 71.8 | 72.9 | 77.8 |
| *video only* | *Human* | 69.1 | 64.3 | 86.4 | 73.3 | 58.5 | 60.0 | 76.7 | 63.3 | 64.6 | 68.9 |
| | Video-Llama2-13B | 24.0 | 32.0 | 67.0 | 41.0 | 50.7 | 45.3 | **56.0** | 52.0 | 51.0 | 46.0 |
| | LLaVA-7B | 33.0 | 15.0 | 69.0 | 39.0 | 44.0 | 24.0 | **56.0** | 57.3 | 45.3 | 42.2 |
| | GPT-4V | 64.0 | 34.0 | 39.0 | 45.7 | 54.7 | 26.7 | 48.0 | 56.0 | 46.3 | 46.0 |
| | BIPALM *w/* GPT-J-6B | 63.0 | 57.0 | **72.0** | **64.0** | 45.3 | **62.7** | 50.7 | **62.7** | 55.3 | 59.7 |
| | BIPALM *w/* Llama2-7B | **69.0** | **63.0** | 60.0 | **64.0** | **62.7** | 54.7 | 53.3 | **62.7** | **58.3** | **61.2** |
| *multimodal* | *Human* | 95.8 | 96.7 | 100 | 97.5 | 90.0 | 91.7 | 83.3 | 88.9 | 88.5 | 93.0 |
| | Video-Llama2-13B | 36.0 | 38.0 | 52.0 | 42.0 | 36.0 | 41.3 | 30.7 | 45.3 | 38.3 | 40.2 |
| | LLaVA-7B | 46.0 | 14.0 | 69.0 | 43.0 | 65.3 | 22.7 | 40.0 | 48.0 | 44.0 | 43.5 |
| | GPT-4V | **94.0** | 13.0 | 59.0 | 55.3 | 56.0 | 26.7 | 4.0 | 52.0 | 34.7 | 44.0 |
| | BIPALM *w/* GPT-J-6B | 90.0 | 69.0 | 86.0 | 81.7 | 68.0 | 78.7 | 56.0 | 73.3 | 69.0 | 75.3 |
| | BIPALM *w/* Llama2-7B | 88.0 | 68.0 | 85.0 | 80.3 | 62.7 | 77.3 | 72.0 | 80.0 | 73.3 | 76.7 |
| | **Ours (*w/* Llama3.1-405B)** | 92.1 | **76.0** | **93.0** | **87.1** | **73.4** | **80.0** | **75.5** | 78.7 | **76.9** | **81.3** |

when controlling the 70B large LM, demonstrating its superior transferability. These results highlight two key points: ***(i) downsizing the weak component can still effectively guide larger LMs without a significant loss in accuracy, and (ii) reducing model width, rather than depth, tends to be more generalizable, as deeper models demonstrate better transferability***—aligning with learning principles of the width-depth trade-offs in small-scale studies (Telgarsky, 2016; Lu et al., 2017; Raghu et al., 2017).

### 4.5. Transferability of scaled Bayesian planner

Although the small LMs are post-trained on the *apartment*, our overall framework is expected to be stable and generalizable across various unseen scenarios. To evaluate the transferability, Tab.4 compares our method with baseline models in five previously unseen scenarios: *Andersen fairy tales, ancient Egyptian, outer space, wild west, and medieval castle*. These diverse settings assess the generalisability of our approach beyond the post-training scenario. When scaling the strong component (i.e., the large controlled LMs) from 70B to 405B across these new scenarios, there are continuous improvements in ToM understanding. **This demonstrates that the increased capacity of our scaled solution enhances the transferability of ToM reasoning across multiple dynamic and unseen environments.** Furthermore, when the weak controller component is reduced from 8B to 4B, performance remains stable, ranging between 78.0% and 79.15%. This result is comparable to the 79.05% accuracy

achieved in the original apartment scenario and also remains close to the performance of the 8B LMs. This consistency suggests that downsizing the weak component does not significantly affect performance, even in new and diverse test environments. **These results indicate that our approach has strong potential for continually downsizing smaller LMs as controllers since they also are capable of capturing the post-trained behaviours.** It allows saved resources to be potentially allocated to stronger controlled LMs, while still keeping stable to scenarios unseen previously.

### 4.6. Weak-to-strong control redirects large LM

To quantify the influence of the weak controller in Bayesian planner, we analyze the estimated likelihood changes before and after applying weak-to-strong control at each step. Fig.3 samples ten test cases from five datasets and averages the results. It illustrates the progressively increasing magnitude of likelihood changes as Bayesian inference progresses:

*(i)* At the beginning, when the large LM is close to a general initial state, the likelihood changes are minimal. This is because the general state aligns closely with pretrained world knowledge, requiring little correction from ToM-specific behaviours; *(ii)* As the model approaches a more specialized final hypothesis, the likelihood estimates are increasingly redirected. This occurs because the specialized scenarios demand ToM-specific behaviours, which the post-trained small LMs are fine-tuned to capture. The post-trained small

*Table 2.* Scaling-up performance on strong component (large LMs) in weak-to-strong control.

| LM | config | belief inference | | | | goal inference | | | | | all |
|---|---|---|---|---|---|---|---|---|---|---|---|
| | | 1.1 | 1.2 | 1.3 | avg. | 2.1 | 2.2 | 2.3 | 2.4 | avg. | |
| Llama2 | 7B-zero-shot | 44.00 | 37.00 | 84.00 | 55.00 | 64.00 | 65.33 | 62.67 | 64.00 | 64.00 | 60.14 |
| | 7B-post-trained | 80.00 | 60.00 | 89.00 | 76.33 | 74.67 | 60.00 | **78.67** | 66.67 | 70.00 | 72.71 |
| | 70B-zero-shot | 64.00 | 47.00 | **93.00** | 68.00 | 56.00 | 72.00 | 25.33 | 70.67 | 56.00 | 61.14 |
| | 70B-post-trained | **90.00** | **70.00** | 87.00 | 82.33 | **78.67** | **76.00** | 61.33 | 72.00 | 72.00 | 76.43 |
| | **70B-ours** | 89.00 | **70.00** | 90.00 | **83.00** | 73.33 | 74.67 | 76.00 | **73.33** | 74.33 | **78.05** |
| Llama3 | 8B-zero-shot | 88.00 | 72.00 | 91.00 | 83.67 | 65.33 | 57.33 | 13.33 | 53.33 | 47.33 | 62.90 |
| | 8B-post-trained | **92.00** | 72.00 | 83.00 | 82.33 | **77.33** | 73.33 | 72.00 | 70.67 | 73.33 | 77.19 |
| | 70B-zero-shot | 69.00 | 67.00 | **95.00** | 77.00 | 42.67 | 70.67 | 16.00 | 52.00 | 45.33 | 58.90 |
| | 70B-post-trained | 91.00 | 70.00 | 89.00 | 83.33 | 73.33 | **74.67** | 44.00 | 69.33 | 65.33 | 73.05 |
| | **70B-ours** | 91.00 | **75.00** | 92.00 | **86.00** | 68.00 | 72.00 | **74.67** | **78.67** | 73.33 | **78.76** |
| Llama3.1 | 8B-zero-shot | 88.00 | 72.00 | 91.00 | 83.67 | 65.33 | 62.67 | 22.67 | 54.67 | 51.33 | 65.19 |
| | 8B-post-trained | 90.00 | 71.00 | 93.00 | 84.67 | 69.33 | 72.00 | 62.67 | 72.00 | 69.00 | 75.71 |
| | 70B-zero-shot | 85.00 | 63.00 | 93.00 | 80.33 | 72.00 | 76.00 | 16.00 | 61.33 | 56.33 | 66.62 |
| | 70B-post-trained | 91.00 | 69.00 | **95.00** | 85.00 | 69.33 | 80.00 | 29.33 | 69.33 | 62.00 | 71.86 |
| | 405B-zero-shot | 86.00 | 70.00 | 90.00 | 82.00 | 73.33 | 78.67 | 21.33 | 66.67 | 60.00 | 69.43 |
| | **70B-ours** | 90.00 | 74.00 | 93.00 | 85.67 | **74.67** | 77.33 | 70.67 | 76.00 | 74.67 | 79.38 |
| | **405B-ours** | **92.10** | 76.00 | 93.00 | **87.10** | 73.40 | 80.00 | 76.50 | 78.67 | 77.14 | **81.29** |
| 3.3 | 70B-post-trained | 91.20 | 71.21 | 94.10 | 85.50 | 74.50 | 79.43 | 66.50 | 79.00 | 74.86 | 80.18 |
| | **70B-ours** | 92.33 | 72.00 | 94.00 | 86.11 | 75.20 | 81.10 | 75.00 | 77.85 | 77.79 | 81.95 |

*Table 3.* Scaling-down effect on weak part (small LMs) in scaled Bayesian planning. All models are based on Llama3.1.

| LM | config | belief inference | | | | goal inference | | | | | all |
|---|---|---|---|---|---|---|---|---|---|---|---|
| | | 1.1 | 1.2 | 1.3 | avg. | 2.1 | 2.2 | 2.3 | 2.4 | avg. | |
| 8B | zero-shot | 88.00 | 72.00 | 91.00 | 83.67 | 65.33 | 62.67 | 22.67 | 54.67 | 51.33 | 65.19 |
| | post-trained | 90.00 | 71.00 | 93.00 | 84.67 | 69.33 | 72.00 | 62.67 | 72.00 | 69.00 | 75.71 |
| | **8B ↝ 70B** | **90.00** | **74.00** | **93.00** | **85.67** | **74.67** | **77.33** | **70.67** | **76.00** | **74.67** | **79.38** |
| 4B wid. | zero-shot | 79.00 | 69.00 | 89.00 | 79.00 | 60.00 | 69.33 | 24.00 | 52.00 | 51.33 | 63.19 |
| | post-trained | 90.00 | 72.00 | 87.00 | 83.00 | 70.67 | 72.00 | 68.00 | **78.67** | 72.33 | 76.90 |
| | **4B-width ↝ 70B** | **90.00** | 71.00 | **90.00** | **83.67** | **74.67** | **74.67** | **76.00** | 73.33 | **74.67** | **78.52** |
| 4B dep. | zero-shot | 91.00 | **74.00** | 88.00 | 84.33 | 69.33 | **77.33** | 20.00 | 66.67 | 58.33 | 69.48 |
| | post-trained | 91.00 | 71.00 | 90.00 | 84.00 | 65.33 | 65.33 | 76.00 | **69.33** | 69.00 | 75.43 |
| | **4B-depth ↝ 70B** | **91.00** | 72.00 | **91.00** | **84.67** | **72.00** | 74.67 | **84.00** | 64.00 | **73.67** | **78.38** |

LMs are specifically fine-tuned to the ToM context, enabling them to model human actions, goals, beliefs, and environmental states across unseen scenarios. **Overall, this analysis finds that the weak component progressively redirects the output of larger models, guiding them toward more accurate ToM predictions among unseen scenarios throughout the Bayesian inference.**

## 4.7. Post-training aligns large LM's likelihood estimation at the concept level

Previous experiments demonstrated that post-training on small LMs can progressively guide the behaviour of large LMs throughout Bayesian inference. Now, we further focus on how post-trained small LMs influence large LMs'

likelihood estimation **at the concept level**. Fig.4 shows the execution of ten inference trials with a temperature of 0.7. The scenario involves the agent James interacting with objects in an apartment, aiming to retrieve a bottle of wine. The initial state $s_i$ is *pear in the basket, no wine*, the belief $b_i$ is *wine in the cabinet*, the goal $g_i$ is *obtain a bottle of wine*, and the action $a_i$ is *open basket, walk to cabinet*. The baseline small LM assigns lower likelihoods to fine-grained item-level concepts (e.g., wine, wine glass). After post-training, the small LM significantly shifts its focus toward item-level concepts, aligning its predictions more closely with the action space. This adjustment increases the likelihood assigned to critical items like wine and wine glass, which are necessary for accurately predicting the agent's

*Table 4.* Transfer performance of the Bayesian method with different scaling settings (zero-shot, direct post-training, and our weak-to-strong control) from the apartment scenario to various unseen environments. All models are based on Llama3.1. Results are average accuracy of *belief inference/goal inference/overall* for each scenario. Detailed unseen scenarios and results are in §E.6&E.7.

| | solution | apartment *(seen)* | Andersen tales | ancient Egyptian | outer space | wild west | medieval castle |
|---|---|---|---|---|---|---|---|
| **Raw** | 70B-zero-shot | 80.3/56.3/66.6 | 83.6/60.6/70.2 | 83.6/60.6/69.3 | 84.0/58.0/69.1 | 82.6/57.6/68.3 | 82.6/57.6/68.3 |
| | 70B-post-trained | 85.0/62.0/71.8 | 84.6/66.3/74.1 | 84.6/66.3/75.3 | 83.0/66.0/73.2 | 81.0/65.0/71.8 | 81.0/65.0/71.8 |
| **Ours** | **4B-wide ↝ 70B** | 83.6/74.6/78.5 | 84.0/75.3/79.0 | 83.0/75.3/79.1 | 82.6/75.3/78.4 | 84.0/74.6/78.6 | 84.6/73.0/78.0 |
| | **4B-depth ↝ 70B** | 84.6/73.6/78.3 | 85.0/71.3/77.1 | 85.3/71.3/77.9 | 81.6/71.0/75.5 | 83.3/71.3/76.4 | 83.3/64.0/72.2 |
| | **8B ↝ 70B** | 85.6/74.6/79.3 | 82.6/76.0/78.8 | 83.6/76.0/77.7 | 84.0/75.0/78.8 | 83.3/74.0/78.0 | 83.6/75.0/78.7 |
| | **8B ↝ 405B** | 87.0/77.0/81.3 | 85.8/76.0/80.2 | 86.0/76.3/80.4 | 87.2/75.5/80.5 | 85.3/76.0/79.9 | 85.6/75.2/79.7 |

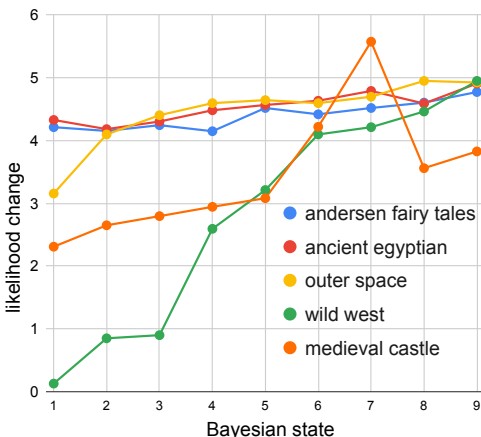

*Figure 3.* Likelihood change during Bayesian inference under weak-to-strong control. Results are averaged over ten sampled cases across five different unseen scenarios.

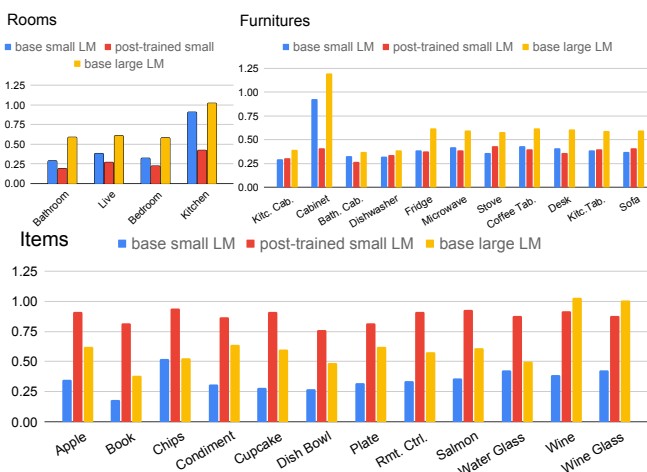

*Figure 4.* Likelihood estimation across different levels of concept granularity (rooms, furniture, and items) for base small LM, post-trained small LM, and base large LM. The Bayesian framework uses an LM as the **policy model** to infer actions conditioned on states, beliefs, and goals, where actions often refer to fine-grained item-level concepts (e.g., wine, wine glass). It highlights the trend of how each model allocates likelihood across these concept levels. The ToM scenario of this case is detailed at §E.5.

*Table 5.* Directly adding a small LM (8B) to a large LM by naive logit combination, without weak-to-strong control, is suboptimal.

| LM | Method | Belief Avg. | Goal Avg. | All Avg. |
|---|---|---|---|---|
| Llama3.1 70B | Naive Logit Add | 82.50 | 66.40 | 74.45 |
| Llama3.1 405B | Naive Logit Add | 83.67 | 65.67 | 74.67 |

goal. Consequently, post-training enables the small LM to better capture fine-grained details of the agent's behaviour, improving ToM predictions. In contrast, the large LM distributes its likelihood more evenly across all levels, from rooms to items, reflecting a broad understanding of the environment. While this approach captures general spatial awareness—identifying key areas like the kitchen and furniture like the cabinet—it lacks the sharp focus on fine-grained details, such as *wine* and *wine glass*, which are crucial for this task. As a result, the large LM may struggle with tasks that require precise, item-level predictions.

Overall, post-training helps small LMs focus on item-level concepts, making it more effective for this task. While the large LM captures a broader understanding of the physical environments, it benefits from post-trained behaviours that redirect its likelihood estimation toward fine-grained, item-level predictions. **This finding reflects the role of post-trained small LMs in guiding large LMs' concepts in ToM reasoning.**

**Is Simply Adding a Small LM Effective?** We also examine whether naïvely combining a small LM with a large LM—by directly adding their logits, without structured W2S adjustment—can close the gap. As shown in Tab.5, this approach yields lower performance than our method. The results demonstrate that naïvely incorporating a small LM is suboptimal; explicit weak-to-strong control is essential to effectively abstract the specialized ToM behaviors of the small LM and fuse them with the large LM's pretrained world knowledge. These ablation studies conclusively validate that our weak-to-strong control mechanism is both critical and independently responsible for the strong generalization and ToM grounding observed in our method.

## 5. Related Work

### 5.1. Modelling human mental states

There are many studies on understanding human behaviour by classifying and predicting physical motion patterns (Aggarwal & Ryoo, 2011; Caba Heilbron et al., 2015; Choi & Savarese, 2013; Shu et al., 2015). Beyond physical behaviour, some studies focus specifically on modeling human mental states, i.e. ToM. ToM models have followed two broad approaches: Bayesian methods and end-to-end deep learning. Bayesian ToM models (Baker et al., 2017; Jara-Ettinger, 2019; Shu et al., 2021) rely on structured probabilistic frameworks to infer mental states from sparse observations of human behaviour. On the other hand, end-to-end models such as ToMnet (Rabinowitz et al., 2018; Shu et al., 2021; Sclar et al., 2022) have been trained directly on ToM tasks, learning relationships between data patterns without explicit causal models of mental states (Sap et al., 2022; Zhi-Xuan et al., 2022; Ullman, 2023). More recently, neurosymbolic reasoning systems use the neural models for feature extraction, while also incorporating probabilistic models for structured reasoning (Wong et al., 2023; Ying et al., 2024; 2023). They face challenges in dynamic and multimodal environments, where both physical and mental state reasoning are required. Different from prior studies, our work operates in more complex and dynamic multimodal ToM environments, where physical actions and mental state reasoning are intertwined.

### 5.2. Post-training LMs for downstream tasks

Post-training can project the LMs' pre-trained capabilities into downstream tasks such as dialogue generation (Ouyang et al., 2022), multilingual understanding (Yang et al., 2025b;a), prosocial alignment (Bai et al., 2022; Liu et al., 2024b), calibration (Fu et al., 2025), and multimodal tasks (OpenAI, 2023b; Liu et al., 2023; Jian et al., 2023; Liu et al., 2025a; Jian et al., 2024; Diao et al., 2024; Zhang et al., 2025a; Diao et al., 2025b;a; Liu et al., 2025b). Previous approaches also use activation engineering/vector steering to adjust the output predictions of fine-tuned LMs, interpolating the effects of fine-tuning with pre-trained knowledge for diverse downstream tasks (Liu et al., 2021; Mitchell et al., 2024; Liu et al., 2024a; Tan et al., 2024; Cao et al., 2024). Recently, LLMs are post-trained as action/policy models for decision-making in embodied agents, allowing them to interact with and explore environments (Kim et al., 2024; Szot et al., 2024; Li et al., 2024). Our study differs by framing LMs as **policy models** in the context of Bayesian inverse inference, specifically to model human mental states. We address the limitations of existing ToM methods by scaling large policy models at test time using a likelihood redirection strategy, reasoning more accurately in complex ToM scenarios. See App.A.1 for additional discussions.

## 6. Discussion and Conclusion

This study investigates scalable Bayesian inference in complex and dynamic ToM environments. Existing methods based on normal-sized LMs often fail to provide sufficient reasoning capabilities and world knowledge, particularly when used as likelihood estimators in diverse challenging ToM scenarios. Therefore, to overcome these limitations, our solution abstracts and transfers the post-trained behavioral patterns of smaller LMs. This approach allows the extensive world knowledge of large LMs to be progressively redirected towards ToM reasoning at inference time. Consequently, we avoid additional post-training resources for large models, yet allow effective inference-time scaling of Bayesian ToM reasoning even in dynamic and complex physical scenarios.

## Impact Statement

This study advances Bayesian ToM inference *at scale* and contributes new datasets representing diverse cultural contexts. These datasets are based on ancient or fictional cultures, mitigating potential sensitivities related to contemporary societal issues. To ensure ethical integrity, the datasets have been carefully reviewed to minimize concerns regarding discrimination, bias, and fairness. They do not contain real individuals, eliminating risks to privacy or security.

We are committed to responsible research practices and encourage ongoing scrutiny of potential biases or unintended consequences. Our methods are designed to be fully reproducible, with detailed descriptions of datasets, experimental settings, and methodologies provided in this paper and our repository: `https://github.com/chunhuizng/scale-bayesian-planner`.

## Author Contributions

**Student Authors**   Chunhui Zhang led the conceptual development, implementation, figure design, manuscript writing, and preparation of the rebuttal. Zhongyu Ouyang conducted additional experiments, contributed to figure design, refined the manuscript, and assisted with the rebuttal.

**Senior Authors**   Shao-Yuan Lo supervised Chunhui Zhang during the internship phase of this project. Soroush Vosoughi supervised the overall publication process and contributed to idea generation in his capacity as Chunhui Zhang's PhD advisor. Sean Dae Houlihan, a postdoctoral fellow working with Soroush Vosoughi, provided expertise in cognitive science. Kwonjoon Lee and Nakul Agarwal provided valuable discussions and feedback throughout the internship period.

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

# A. Comparison of Methodologies for ToM Inference

Table 6. Attributes of each method for ToM task.

| method | scalability | structured reasoning | world knowledge | multimodality |
|---|---|---|---|---|
| Bayesian ToM models | ✗ | ✓ | ✗ | ✗ |
| end-to-end ToM models | ✗ | ✗ | ✓ | ✓ |
| **ours** | ✓ | ✓ | ✓ | ✓ |

Tab.6 provides a comparative analysis of various methodologies for ToM inference, supplementing the discussion in the introduction (§1). Our proposed approach differs significantly from the underlying philosophies of Bayesian ToM models and end-to-end models. While Bayesian models emphasize structured reasoning guided by principles from cognitive science, they often lack scalability and struggle to handle multimodal inputs. In contrast, end-to-end models incorporate extensive world knowledge but lack the structured reasoning capabilities essential for accurate ToM inference.

Our method integrates these attributes: scalability (e.g., up to 405B), structured reasoning, world knowledge, and the ability to process multimodal inputs. Furthermore, our method demonstrates superior scalability, leveraging the stronger reasoning capabilities of large LMs at test time without the need for extensive post-training on large models. This allows our approach to efficiently handle complex and dynamic ToM scenarios.

## A.1. Our Theoretical Rationales in Scaled ToM Inference and Related Work

Our approach is based on a high-level principle derived from Theorem 1 and its proof, which implies that smaller models can approximate the scaled gradient of the loss function for larger models. This mechanism bypasses direct parameter updates in the larger model, capturing the primary adjustments needed for fine-tuning while exploiting the innate generalisation capacity of the larger model. By relying on the approximate knowledge provided by the smaller model, our framework reduces computational overhead and improves scalability.

This principle is related with previous studies that have explored reweighting mechanisms for various applications (where not necessarily the same as our perspective of scaling or embodied policy model), including avoiding toxicity in text generation (Liu et al., 2021), mitigating harmful outputs in aligned models (Zhou et al., 2024), adjusting code generation (Mitchell et al., 2024), controlling sentiment in text (Han et al., 2024), and reducing hallucination or degeneration in neural text (Chuang et al., 2024; Su et al., 2022). These works demonstrate how reweighting can approximate the behaviour of large language models, mimicking direct fine-tuning in specific contexts. In contrast, our scaled ToM inference extends this principle beyond text generation

tasks into the domain of social cognitive reasoning. Our framework uses language models to approximate policy behaviours for probability estimation in embodied simulators, based on the cognitive science-inspired Bayesian ToM framework. Unlike previous work focusing on text-based tasks such as sentiment or factuality control, our method addresses the unique challenges of ToM tasks, which require complex reasoning and the integration of world knowledge. These tasks involve multimodal scenarios that require understanding of agents' beliefs, goals and actions - a domain distinct from the text generation problems addressed in previous studies.

# B. Data Flow and Processing in Scalable Bayesian ToM Inference

## B.1. Overall Data Flow

For a detailed depiction of the data flow in our method, refer to Fig.5. The symbolic representation tools first convert video and textual descriptions into structured symbolic inputs, which are then processed by the Bayesian inference framework. This framework leverages a large LM as a scaled policy model, dynamically controlled by task-specific priors provided by a post-trained small LM, enabling accurate estimation of action likelihoods in dynamic scenarios.

## B.2. Data Preprocessing: Unified Symbolic Representations

To enable Bayesian ToM inference at scale, following established methods mentioned in MMToM (Jin et al., 2024; Blukis et al., 2022), multimodal data (video and textual descriptions) are transformed into structured symbolic representations. This process involves three key components: **visual perception**, **text parsing**, and **information fusion**. Together, these components provide a unified representation of states, actions, and hypotheses required for ToM tasks.

**Visual Perception.** The visual perception module is designed to process video frames and extract symbolic representations of the environment. For each frame, a scene graph is generated to capture the spatial and relational properties of objects and agents with the scene graph generator(Blukis et al., 2022). Following established methods in MMToM (Jin et al., 2024), voxel maps and 3D bounding boxes are utilized to infer object positions, containment relationships, and human poses. For instance, objects such as *pear* and *basket* are represented by predicates like `In(pear, basket)`. These predicates effectively summarize the physical state of the environment, serving as critical inputs for subsequent reasoning steps.

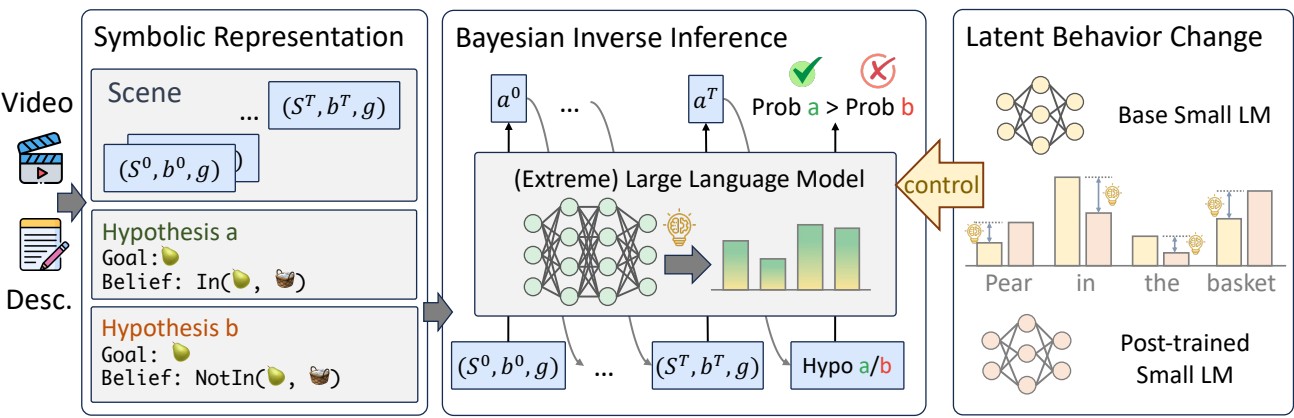

*Figure 5.* The data flow in our scalable Bayesian ToM inference framework. Video scenes and their corresponding descriptions are first processed by multimodal symbolic representation tools (Jin et al., 2024), generating structured symbolic inputs (states, beliefs, goals). These symbolic representations are then integrated into the Bayesian inference process, where a large language model (LM) operates as a scaled **policy model** to estimate the likelihood of an agent's actions in dynamic environments. The right panel demonstrates the latent behavioral changes introduced by the post-trained small LM, which provides task-specific priors to guide the larger LM via a control mechanism.

**Text Parsing.**  To extract symbolic representations from textual descriptions, one LLM (e.g., GPT-4) processes the text into three distinct components: (i) the initial state of the environment, (ii) human actions, and (iii) the question. Each component is translated into symbolic predicates. For example:

- The **state** is represented as predicates like `In(pear, basket)`.

- The **action** is represented as commands such as `walk towards kitchen`.

- The **question** is decomposed into two hypotheses, each comprising a goal (e.g., `pear`) and a belief (e.g., `In(pear, basket)` or its negation, `¬In(pear, basket)`).

This symbolic parsing ensures compatibility with the structured reasoning framework.

**Fusion.**  The fusion module integrates symbolic information from video and text into a unified representation. First, predicates extracted from video inputs (e.g., spatial relationships) are aligned with those parsed from text to form the **initial state**. Next, human actions detected from the video are matched with text-based actions, and the video sequence is segmented into discrete time steps corresponding to these actions. Starting from the initial state, the symbolic representation of the environment is updated at each time step based on newly detected predicates. This process results in a sequence of symbolic states and actions, which serve as the input for Bayesian inference. Additionally, the parsed

question provides two hypotheses—goal and belief—that guide the reasoning task.

## C. Theoretical Rationale

**Theorem 1.** *Let $\pi^{\mathcal{L}}$ be a pretrained base model, $\pi^{\mathcal{E}}$ and $\pi^{\mathcal{N}}$ be smaller tunable models where $\pi^{\mathcal{E}}$ is fine-tuned on the target task, and $\pi^*$ be the directly tuned base model. Suppose the output adjustment $\Delta s(X_t|x_{<t}) = s_{\pi^{\mathcal{E}}}(X_t|x_{<t}) - s_{\pi^{\mathcal{N}}}(X_t|x_{<t})$ approximates the scaled negative gradient of the cross-entropy loss for outputs, i.e., $\Delta s \approx -\eta \nabla_s \mathcal{L}_{CE}(s_{\pi^{\mathcal{L}}}, y)$, where $\eta$ is the learning rate. Then, the proxy-tuned model $\tilde{\pi}$, defined by $s_{\tilde{\pi}}(X_t|x_{<t}) = s_{\pi^{\mathcal{L}}}(X_t|x_{<t}) + \Delta s(X_t|x_{<t})$, approximates the directly tuned base model $\pi^*$. The KL divergence between their output distributions has this relation:*

$$D_{KL}(P_{\pi^*} \| P_{\tilde{\pi}}) \le \frac{\eta^2}{2} \lambda_{\max} \|\nabla_s \mathcal{L}_{CE}(s_{\pi^{\mathcal{L}}}, y)\|_2^2 + \mathcal{O}(\eta^3), \tag{9}$$

*where $\lambda_{\max}$ is the maximum eigenvalue of the Hessian of the cross-entropy loss for the outputs.*

*Proof.* When the learning rate $\eta$ is small, and the cross-entropy loss $\mathcal{L}_{CE}$ is smooth and twice differentiable with respect to the outputs $s$, then the output adjustment $\Delta s$ approximates the scaled negative gradient of the loss as:

$$\Delta s \approx -\eta \nabla_s \mathcal{L}_{CE}(s_{\pi^{\mathcal{L}}}, y). \tag{10}$$

The outputs of the directly tuned base model $\pi^*$ after fine-tuning are updated using gradient descent:

$$s_{\pi^*} = s_{\pi^{\mathcal{L}}} - \eta \nabla_s \mathcal{L}_{CE}(s_{\pi^{\mathcal{L}}}, y) + \frac{\eta^2}{2} H_s(\nabla_s \mathcal{L}_{CE}(s_{\pi^{\mathcal{L}}}, y)) + \mathcal{O}(\eta^3), \tag{11}$$

where $H_s$ is the Hessian of $\mathcal{L}_{\text{CE}}$ with respect to the outputs. The outputs of the proxy-tuned model $\tilde{\pi}$ are:

$$s_{\tilde{\pi}} = s_{\pi^{\mathcal{L}}} + \Delta s. \tag{12}$$

When $\Delta s \approx -\eta \nabla_s \mathcal{L}_{\text{CE}}(s_{\pi^{\mathcal{L}}}, y)$, we have:

$$s_{\tilde{\pi}} \approx s_{\pi^{\mathcal{L}}} - \eta \nabla_s \mathcal{L}_{\text{CE}}(s_{\pi^{\mathcal{L}}}, y). \tag{13}$$

The difference in outputs between the directly tuned model and the proxy-tuned model is:

$$\epsilon_s = s_{\pi^*} - s_{\tilde{\pi}}. \tag{14}$$

Then we consider their expressions:

$$\epsilon_s \approx \frac{\eta^2}{2} H_s(\nabla_s \mathcal{L}_{\text{CE}}(s_{\pi^{\mathcal{L}}}, y)) + \mathcal{O}(\eta^3). \tag{15}$$

The KL divergence between the output distributions of $\pi^*$ and $\tilde{\pi}$ is constrained using the properties of the softmax function and the Lipschitz continuity of the KL divergence:

$$D_{\text{KL}}(P_{\pi^*} \| P_{\tilde{\pi}}) \leq \frac{1}{2} \|\epsilon_s\|_2^2. \tag{16}$$

Using the norm of $\epsilon_s$:

$$\|\epsilon_s\|_2^2 \approx \frac{\eta^4}{4} \|H_s(\nabla_s \mathcal{L}_{\text{CE}}(s_{\pi^{\mathcal{L}}}, y))\|_2^2. \tag{17}$$

The Hessian's norm is constrained by its maximum eigenvalue:

$$\|H_s(\nabla_s \mathcal{L}_{\text{CE}})\|_2 \leq \lambda_{\max} \|\nabla_s \mathcal{L}_{\text{CE}}(s_{\pi^{\mathcal{L}}}, y)\|_2, \tag{18}$$

which gives:

$$\|\epsilon_s\|_2^2 \leq \frac{\eta^4}{4} \lambda_{\max}^2 \|\nabla_s \mathcal{L}_{\text{CE}}(s_{\pi^{\mathcal{L}}}, y)\|_2^2. \tag{19}$$

Finally, the KL divergence is:

$$D_{\text{KL}}(P_{\pi^*} \| P_{\tilde{\pi}}) \leq \frac{\eta^2}{2} \lambda_{\max} \|\nabla_s \mathcal{L}_{\text{CE}}(s_{\pi^{\mathcal{L}}}, y)\|_2^2 + \mathcal{O}(\eta^3). \tag{20}$$

$\square$

For theoretical implications for practical applicability, this analysis demonstrates that the weak-to-strong control mechanism relies on the learned $\Delta s$ to approximate the scaled gradient $-\eta \nabla_s \mathcal{L}_{\text{CE}}(s_{\pi^{\mathcal{L}}}, y)$ with higher-order terms contributing to the residual error. Importantly, our method does not require the small LM ($\pi^{\mathcal{E}}$) to strictly approximate the exact gradient of the cross-entropy loss for the large model. Instead, the large model ($\pi^{\mathcal{L}}$) leverages its intrinsic capacity for generalization and adaptation, based only on the approximate adjustment $\Delta s$ learned by the small LM.

This inherent flexibility allows the large model to harness its pre-trained potential, activated by the weak-to-strong control mechanism, to effectively adapt to the current ToM task. Consequently, our method achieves stable advanced performance even in novel scenarios where the small LM provides only a coarse approximation of the gradient. This significantly reduces the reliance on strict fine-tuning and maximizes computational efficiency, ensuring the approach is both scalable and practical for the physical VirtualHome environment.

## D. Experimental Details

### D.1. Belief and Goal Inference Types and Their Characteristics to LMs

MMToM apartment scenario questions are split into seven types, assessing ToM reasoning (Jin et al., 2024): Belief Inference includes 50% of questions on True Belief (*Type 1.1*), False Belief (*Type 1.2*), and Long-Term Belief Tracking (*Type 1.3*). Goal Inference covers the remaining 50% on True Belief (*Type 2.1*), False Belief (*Type 2.2*), Updated Belief (*Type 2.3*), and Future Actions (*Type 2.4*).

Short-term Belief Inference relies heavily on world knowledge, making it more responsive to enhancements from large LMs' pretrained capabilities. In contrast, long-term reasoning—both for Belief and Goal Inference—focuses on the dynamic nature of the environment and benefits from post-training specifically aligned to ToM scenarios.

### D.2. Post-training configurations

Tab.7 summarizes the LoRA post-training configurations applied to Llama2, Llama3, and Llama3.1 models during policy model training. We carefully adjust $\alpha$, rank, and other hyperparameters to optimize performance across different model sizes. Notably, following prior engineering studies, a higher $\alpha$ and rank are used for smaller models (7B and 8B), while reduced values are employed for the larger 70B model to ensure efficient adaptation without overfitting.

#### D.2.1. FINE-TUNING PROCESS AND RESOURCES

The fine-tuning process for smaller models (e.g., Llama3.1-8B) was conducted using a single NVIDIA H100 GPU, leveraging BF16 mode to optimize memory usage and maintain GPU memory consumption under 60GB. This configuration enabled efficient training of policy models tailored for Theory of Mind (ToM) tasks. The fine-tuning process was executed with the following parameters:

- **Batch size:** 16 (achieved via a per-device batch size of 4 and gradient accumulation steps of 4),

- **Learning rate:** $5 \times 10^{-5}$,

- **Number of epochs:** 3.

Under this setup, the fine-tuning process required approximately 8 hours to converge.

### D.2.2. DATASET SIZE

The training pool size $N$ for post-training was set to 20,000 data points, sourced from the MMToM dataset's training split and our released data sampled from an embodied simulator. For tasks involving transfer to new themes, the training dataset size remained consistent at 20,000 data points, ensuring a fair and uniform setup across different experiments.

## E. Additional Experiments

### E.1. Comparison of Fine-Tuning Methods on MMToM Tasks

To evaluate the relative performance of full fine-tuning (FFT) and LoRA fine-tuning, we conducted experiments on two smaller models, GPT2-large (Radford et al., 2019) (774M parameters) and Gemma-2B (2B parameters) (Team et al., 2024). Each model was fine-tuned using datasets of 20,000 and 8,000 datapoints, over two epochs, on 8 NVIDIA A100 80GB GPUs. The results are summarised in Tab.8.

The results show several important trends. *First,* when sufficient training data is available (e.g., 20,000 data points), full fine-tuning consistently outperforms LoRA, with accuracy gains of 0.9-1.2 percentage points. This suggests that full training is better at exploiting richer data, especially for smaller models. *Second,* the performance gap between FFT and LoRA narrows for larger models. For example, Gemma-2B shows minimal differences between FFT and LoRA (0.3 percentage points on 20,000 data points), suggesting that larger models are stable to LoRA's parameter efficiency constraints. *Finally,* the influence of dataset size is evident: while FFT shows greater improvements over LoRA on smaller datasets, LoRA maintains competitive performance in resource-constrained scenarios, especially for larger models. Tab.9 further demonstrates the stable performance of weak-to-strong control when transferring ToM-specific fine-tuning knowledge from a smaller model (Minitron-4B-Width) to a larger model (Llama3.1-70B). The difference in accuracy between FFT and LoRA is only 0.15 percentage points when weak-to-strong control is applied, indicating that the mechanism is highly effective at bridging the gap between fine-tuning methods. Importantly, this highlights the ability of the proposed method to scale ToM-specific behaviors efficiently, leveraging both computationally intensive FFT and parameter-efficient LoRA.

Overall, these experiments highlight a trade-off between computational efficiency and performance gains. Full fine-tuning achieves modest but consistent improvements, particularly for smaller models and larger datasets. However, for larger models, LoRA provides an effective alternative with near-parity in performance and significantly reduced computational overhead. Furthermore, our weak-to-strong control mechanism demonstrates stability to fine-tuning methods, enabling scalable ToM-specific behavior elicitation with high accuracy in larger models.

### E.2. Impact of Pre-Training Quality on MMToM Tasks

The differences in performance between the Llama2, Llama3 and Llama3.1 models provide insight into the role of pre-training quality, especially at large model scales. Based on the experimental results in Tab.2 and Tab.3, the influence of pre-training quality diminishes primarily due to a ceiling effect, but this is only observed when comparing models within the same scale, such as the 70B parameter range. However, when comparing smaller models to larger ones, the effect of pre-training is more pronounced. For example, moving from Llama2 7B to Llama2 70B after ToM-specific post-training leads to a 6% improvement in belief inference accuracy (from 76.33% to 82.33%) and a 2% improvement in goal inference accuracy (from 70% to 72%), highlighting the role of scaling in encoding richer representations.

When examining why pre-training becomes less effective at larger scales, such as comparing Llama2-70B (pre-trained with 2.2 trillion tokens) to Llama3.1-70B (pre-trained with 15 trillion tokens), the results suggest that larger pre-training corpora improve performance primarily for tasks that rely heavily on world knowledge: Tasks involving belief inference, which rely on short-term reasoning and general world knowledge, show significant improvements due to improved representations learned during pre-training. For example, Llama3.1 achieves a 3.67% improvement in belief inference accuracy over Llama2 (from 83.00% to 85.67%). These tasks benefit from richer pre-training datasets that refine the model's understanding of common human behaviours and object interactions.

In contrast, goal inference tasks that rely on long-term reasoning, including integrating temporal observations and dynamically updating beliefs, show smaller gains from larger pre-training corpora. For example, Llama3.1 improves goal inference accuracy by only 1.67% over Llama2 (from 72.33% to 74.00%). Such tasks are more dependent on the fine-tuning stage and the use of task-specific reasoning frameworks, such as weak-to-strong control. These results suggest that for complex reasoning tasks, the primary performance bottleneck shifts from pre-training quality to the reasoning strategies employed during fine-tuning.

In summary, pre-training quality has a significant impact on smaller models and tasks that rely heavily on world knowledge, such as belief inference. However, as models scale up to 70B parameters, the influence of pre-training

diminishes due to ceiling effects, and logical reasoning tasks such as goal inference rely more on task-specific adaptations during fine-tuning.

### E.3. How Consistent is Theory of Mind Across Different Phrasings?

As shown in Tab.4, the "*All*" column across different themes (e.g. Apartment, Andersen Fairy Tales, etc.), there is noticeable performance variance even within models of the same scale. To quantify this, we measured the range of variance for three configurations: **70B-zero-shot**, **70B-post-trained** and **8B ↬ 70B**: (1) For 70B-zero-shot, performance ranged from 66.62 to 70.52 across themes, yielding a variance range of **3.90**; (2) For 70B-post-trained, the variance range of post-trained LMs is **3.47**, with performance ranging from 71. 86% and 75.33%; (3) For our solution 8B ↬ 70B, the weak-to-strong control mechanism further stabilised the performance, reaching only the smallest variance range of **1.62**, with scores between 77.76% and 79.38%.

These results suggest that specific topics have different effects on ToM skills, but our solution demonstrates relative stability to distributional changes caused by topic shifts. For example, **70B-zero-shot** achieves its highest performance up to 70.52% and its lowest up to 66.62%, highlighting the model's pronounced sensitivity to thematic variations in reasoning trajectories without adaptation. In contrast, our proposed solution, **8B ↬ 70B**, significantly reduces this gap, demonstrating the effectiveness of the weak-to-strong control mechanism in adjusting the ToM behaviour of the larger model while preserving the framework's general reasoning capacity across diverse and scenario-agnostic contexts.

### E.4. On the Role of the Weak-to-Strong Framework

The weak-to-strong framework presented in this paper focuses on aligning the larger model's distribution with ToM-specific beliefs and task structures while preserving its general reasoning capabilities, rather than primarily relying on the smaller model's reasoning abilities. This design enables efficient transfer of ToM-specific task structures without compromising the broader capabilities of the larger model.

The smaller model (e.g., 4B or 8B parameters) undergoes ToM-specific post-training to encode task-relevant priors, such as belief states and potential goals, without requiring advanced independent reasoning capabilities. During inference, the smaller model functions as an assistive scaffold, conditioning the larger model's likelihood estimation in a Bayesian framework. This role is formalized through the adjustment ratio: $\frac{\pi^{\mathcal{E}}}{\pi^{\mathcal{N}}}$, where $\pi^{\mathcal{E}}$ is the post-trained smaller model's task-specific policy, and $\pi^{\mathcal{N}}$ is the naive pre-trained smaller model's policy.

The larger model (e.g., 70B parameters) integrates this ad-

justment ratio to refine its likelihood estimation dynamically. The overall policy distribution is computed as $\pi^{\mathcal{L}} \frac{\pi^{\mathcal{E}}}{\pi^{\mathcal{N}}}$, where $\pi^{\mathcal{L}}$ is the policy from the larger model. This mechanism allows the larger model to retain its broad reasoning and world knowledge, ensuring its capacity for generalization while aligning with ToM-specific task structures.

To validate this framework, we compared the performance of the 8B ↬ 70B model to the 70B-post-trained model across five unseen themes, including *Andersen Fairy Tales*, *Ancient Egyptian*, and *Outer Space*. As shown in Tab.10, the weak-to-strong mechanism achieved consistent improvements across all ToM tasks, demonstrating its ability to preserve and transfer the larger model's general reasoning capabilities while aligning with ToM-specific requirements. These results, combined with theoretical insights from Section C, demonstrate that the weak-to-strong framework effectively utilizes the smaller model as a task-specific lens to guide the larger model's predictions. This collaborative dynamic ensures alignment with ToM-specific task requirements while preserving general reasoning capabilities.

### E.5. Theory-of-Mind Case Study: Agent James in Apartment Interaction

Fig.6 provides a detailed visual and language-based description of the test case described in experiment §4.7 of the experiment, where the likelihood estimation behaviour of different LMs is discussed across varying concept levels.

### E.6. ToM Transfer Effect on unseen scenarios

Tab.11 supplements the results in experiment §4.5, providing a detailed comparison between the baselines and our scalable solution across belief inference and goal inference subtasks in various unseen ToM scenarios. Our experimental observations are consistent with those outlined in §4.5: *(i)* The increased capacity of our scalable solution significantly improves the transferability of ToM reasoning across dynamic and previously unseen environments. *(ii)* Our approach demonstrates strong potential for downsizing small LMs as controllers, as they successfully capture the post-trained behaviours and exhibit stable performance in guiding larger models. *(iii)* Notably, our method can approximate—and in some cases outperform—the results achieved by directly post-training large-scale LMs (such as the 70B model). These findings underscore the flexibility and scalability of our approach for handling practical ToM tasks in diverse, complex environments.

### E.7. Thematic Scenario Data for ToM Task Transfer

As described in §E.6, five new thematic scenarios are used for evaluation: Andersen Fairy Tales, Ancient Egyptian, Wild West, Outer Space, and Medieval Castle. These en-

vironments are not seen during the post-training phase of our method and are different from the original *apartment* setting.

The transfer to these scenarios demonstrates the generalisability of our solution to dynamically adapt to different domains, with each thematic environment presenting unique challenges and contextual shifts from the apartment scenario. Fig.7 provides a visual summary of these key differences, statistically extracted and mapped to illustrate the transformation of concept and environment across these themes. These distinctions are used to evaluate ToM task transfer across different dynamic environments.

### E.8. Generalization to Diverse Real-world Social Interactions

In Tab.12, we expand our evaluation using MuMA-ToM (Shi et al., 2025), a benchmark explicitly designed for nuanced social interaction, including: *(1) Belief inference:* Understanding environmental dynamics. *(2) Social Goal inference:* Interpreting subtle social objectives. *(3) Belief-of-Goal inference:* Attributing complex mental states. Results in Table A show that our method performs competitively with the state-of-the-art GPT-4o-based LIMP (Shi et al., 2025) and outperforms all the other baselines. Note that this is achieved by using open-source models, avoiding the expensive GPT-4o API cost required by LIMP (Shi et al., 2025). Our weak-to-strong control leverages large pretrained LMs, effectively adapting to real-world social reasoning without compromising generalization.

### E.9. Comparison with Parameter-Efficient Fine-Tuning

The proposed weak-to-strong control is fully orthogonal and complementary to PEFT techniques, i.e., we can combine our method with any PEFT technique. In fact, our small LMs are trained by LoRA, as described in L191-right. Tab.13 further confirms the consistent effectiveness of our method, regardless of the PEFT choice for small LM.

Directly applying PEFT to large pretrained LMs performs worse than our method. As discussed above, our method avoids fine-tuning large LMs and thus preserves their pretrained mental/world knowledge (Kotha et al., 2024; Mitchell et al., 2025; Zheng et al., 2025), essential for generalization in multimodal ToM tasks.

### E.10. The Necessity of Weak-to-Strong Control

To elucidate the contribution of our Weak-To-Strong mechanism, we conduct two ablation studies: (1) replacing W2S with naïve post-training, and (2) simply adding a small LM (8B) to a large LM without structured control.

Tab.14 presents a comparative ablation between our Weak-to-Strong controlled models and their naïvely post-trained counterparts. Across all model scales and architectures, the absence of Weak-to-Strong guidance results in consistent and often substantial drops in generalization performance. These results underscore that Weak-to-Strong control is critical for maximizing the large LM's ability to leverage its pretrained world and mental-state knowledge, enabling stronger ToM reasoning.

## F. Practicality under Real-Time and Resource Constraints

A key concern in deploying advanced ToM reasoning systems is their feasibility under stringent computational and latency requirements. Our approach addresses this by leveraging a small, post-trained LM (e.g., 4B or 8B) to provide dynamic, parallel guidance to a large, pretrained LM (e.g., 70B or 405B) during inference.

Both LMs can be deployed simultaneously on standard NVIDIA H100 GPUs (80GB, BF16 precision), with the small LM introducing negligible computational overhead. For instance, in the 8B+70B configuration, inference over 600 tasks requires approximately 14–15.5 minutes (1.4–1.55 seconds per question), matching the runtime of an unguided 70B model. This efficiency is attributed to the lightweight nature of the small LM's computations relative to the dominant cost of the large LM, and to efficient parallelization with minimal synchronization (passing only compact likelihood tensors).

Moreover, both the small and large LMs only perform likelihood estimation (i.e., prefilling, typically up to 1024 tokens), which is well supported by contemporary acceleration frameworks such as NVIDIA Dynamo, vLLM (Kwon et al., 2023). These optimizations make our Bayesian ToM planner well-suited to real-time and resource-constrained environments.

Finally, our method circumvents the prohibitive costs of fine-tuning very large LMs. Directly fine-tuning a 405B model would typically require upwards of 50–64 H100 GPUs, which is beyond the reach of most institutions. In contrast, our approach enables effective adaptation by fine-tuning only a small LM (e.g., 8B), reducing hardware requirements to a single H100 GPU, while still achieving superior performance through guided inference. This design makes our approach both scalable and accessible for practical deployment.

*Table 7.* LoRA configuration settings for Llama2, Llama3, and Llama3.1 during post-training for policy models.

| configs | 7B | 8B | 70B |
|---|---|---|---|
| bias | none | none | none |
| fan-in fan-out | false | false | false |
| inference mode | true | true | true |
| LoRA initialization | true | true | true |
| $\alpha$ | 32 | 32 | 16 |
| dropout | 0.05 | 0.05 | 0.05 |
| rank | 16 | 16 | 8 |
| target modules | [q-proj, v-proj] | | |
| task type | causal-lm | | |

*Table 8.* Comparison of full fine-tuning (FFT) and LoRA fine-tuning for GPT2-large and Gemma-2B across different MMToM data sizes.

| | Fine-tuning Method | Data Size | Model Size | Accuracy (%) |
|---|---|---|---|---|
| GPT2-large | FFT | 20,000 | 774M | 63.4 |
| GPT2-large | LoRA | 20,000 | 774M | 62.4 |
| GPT2-large | FFT | 8,000 | 774M | 62.8 |
| GPT2-large | LoRA | 8,000 | 774M | 62.1 |
| Gemma-2B | FFT | 20,000 | 2B | 68.8 |
| Gemma-2B | LoRA | 20,000 | 2B | 68.5 |
| Gemma-2B | FFT | 8,000 | 2B | 67.5 |
| Gemma-2B | LoRA | 8,000 | 2B | 67.3 |

*Table 9.* Comparison of weak-to-strong control for Llama3.1-Minitron-4B-Width and Llama3.1-70B using different fine-tuning methods on the smaller model.

| | Fine-tuning Method | Data Size | Model Size | Accuracy (%) |
|---|---|---|---|---|
| Llama3.1-Minitron-4B-Width | FFT | 20,000 | 4B | 77.00 |
| Llama3.1-Minitron-4B-Width | LoRA | 20,000 | 4B | 76.90 |
| *Weak-to-strong control results:* | | | | |
| 4B-Width ↬ Llama3.1-70B | FFT-trained 4B | 20,000 | 70B | 78.67 |
| 4B-Width ↬ Llama3.1-70B | LoRA-trained 4B | 20,000 | 70B | 78.52 |

*Table 10.* Performance of the 8B ↬ 70B LMs on unseen themes compared to 70B-post-trained/-zero-shot LMs across all ToM tasks.

| Unseen Theme | Scale | 1.1 | 1.2 | 1.3 | Avg. | 2.1 | 2.2 | 2.3 | 2.4 | Avg. | All |
|---|---|---|---|---|---|---|---|---|---|---|---|
| **Andersen Fairy Tales** | 70B-zero-shot | 88.00 | 73.00 | 90.00 | 83.67 | 70.67 | 80.00 | 25.33 | 66.67 | 60.67 | 70.52 |
| | 70B-post-train | 90.00 | 71.00 | 93.00 | 84.67 | 73.33 | 61.33 | 61.33 | 69.33 | 66.33 | 74.19 |
| | 8B ↬ 70B | 92.00 | 71.00 | 85.00 | 82.67 | 82.67 | 76.00 | 68.00 | 77.33 | 76.00 | 78.86 |
| **Ancient Egyptian** | 70B-zero-shot | 89.00 | 71.00 | 91.00 | 83.67 | 74.67 | 74.67 | 25.33 | 60.00 | 58.67 | 69.38 |
| | 70B-post-train | 89.00 | 69.00 | 96.00 | 84.67 | 72.00 | 76.00 | 61.33 | 64.00 | 68.33 | 75.33 |
| | 8B ↬ 70B | 90.00 | 73.00 | 88.00 | 83.67 | 69.33 | 76.00 | 73.33 | 74.67 | 73.33 | 77.76 |
| **Outer Space** | 70B-zero-shot | 88.00 | 72.00 | 92.00 | 84.00 | 72.00 | 64.00 | 25.33 | 70.67 | 58.00 | 69.38 |
| | 70B-post-train | 91.00 | 68.00 | 90.00 | 83.00 | 69.33 | 65.33 | 61.33 | 68.00 | 66.00 | 75.33 |
| | 8B ↬ 70B | 90.00 | 70.00 | 92.00 | 84.00 | 73.33 | 81.33 | 66.67 | 78.67 | 75.00 | 77.76 |

> ### ToM exemplar
>
> *What's inside the apartment:*
> The apartment consists of a bedroom, a bathroom, a living room, and a kitchen. In the bedroom, there is a coffee table with a plate on it. The bathroom houses a cabinet, which is currently empty. The living room is furnished with a cabinet, a coffee table, a sofa, and a desk. The cabinet is filled with two apples, a condiment bottle, three wine glasses, two water glasses, a cupcake, two bags of chips, a remote control, and a bottle of wine. Both a water glass and a wine glass are placed on the coffee table. The kitchen is equipped with a fridge, an oven, a kitchen table, and a microwave. Inside the fridge, there are two apples. The oven contains a salmon. Meanwhile, the microwave houses a salmon and two cupcakes.
> *Actions taken by James:*
> James is in the kitchen. He strides towards the stove, opens it, and then shuts it. He then opens the fridge, closes it, opens the microwave, and closes it as well. Finally, he walks towards the living room and approaches the cabinet.
>
> State Modelling:
> (a) James has been trying to get a bottle of wine. ✅
> (b) James has been trying to get an apple. ❌
>
> $s_i$: apples in fridge, no wine
> $b_i$: wine in the cabinet
> $g_i$: obtain a bottle of wine
> $a_i$: open fridge, walk to cabinet

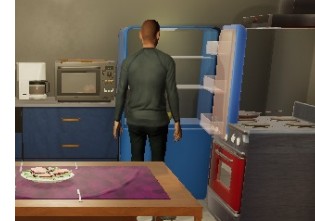
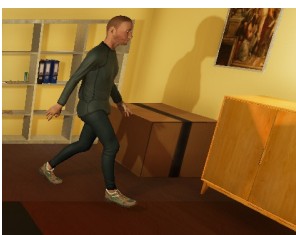

*Figure 6.* Theory-of-Mind scenario used in the main experiments §4.7, involving an agent (James) interacting with objects in an apartment.

*Table 11.* Detailed transfer performance of the Bayesian method with different scaling strategies (zero-shot, direct post-training, and our weak-to-strong control) from the original apartment scenario to various unseen environments. All models are based on Llama3.1.

| Theme | Scale | Belief Inference | | | | Goal Inference | | | | | All |
|---|---|---|---|---|---|---|---|---|---|---|---|
| | | 1.1 | 1.2 | 1.3 | Avg. | 2.1 | 2.2 | 2.3 | 2.4 | Avg. | |
| Andersen fairy tales | 70B-zeroshot | 88.00 | 73.00 | 90.00 | 83.67 | 70.67 | 80.00 | 25.33 | 66.67 | 60.67 | 70.52 |
| | 70B-post-train | 90.00 | 71.00 | **93.00** | 84.67 | 73.33 | 61.33 | 61.33 | 69.33 | 66.33 | 74.19 |
| | **8B ↬ 70B** | **92.00** | 71.00 | 85.00 | 82.67 | **82.67** | 76.00 | 68.00 | **77.33** | **76.00** | **78.86** |
| | **4B-width ↬ 70B** | 90.00 | 73.00 | 89.00 | 84.00 | 80.00 | **81.33** | **76.00** | 64.00 | 75.33 | 79.05 |
| | **4B-depth ↬ 70B** | 91.00 | **74.00** | 90.00 | **85.00** | 74.67 | 73.33 | 64.00 | 73.33 | 71.33 | 77.19 |
| ancient Egyptian | 70B-zeroshot | 89.00 | **71.00** | 91.00 | 83.67 | 74.67 | 74.67 | 25.33 | 60.00 | 58.67 | 69.38 |
| | 70B-post-train | 89.00 | 69.00 | 96.00 | 84.67 | 72.00 | 76.00 | 61.33 | 64.00 | 68.33 | 75.33 |
| | **8B ↬ 70B** | 90.00 | 73.00 | 88.00 | 83.67 | 69.33 | 76.00 | 73.33 | 74.67 | 73.33 | 77.76 |
| | **4B-width ↬ 70B** | 90.00 | 69.00 | 90.00 | 83.00 | 70.67 | **80.00** | **85.33** | 69.33 | **76.33** | **79.19** |
| | **4B-depth ↬ 70B** | **91.00** | 69.00 | **96.00** | **85.33** | **76.00** | 68.00 | 69.33 | **76.00** | 72.33 | 77.90 |
| outer space | 70B-zeroshot | 88.00 | **72.00** | 92.00 | **84.00** | 72.00 | 64.00 | 25.33 | 70.67 | 58.00 | 69.38 |
| | 70B-post-train | **91.00** | 68.00 | 90.00 | 83.00 | 69.33 | 65.33 | 61.33 | 68.00 | 66.00 | 75.33 |
| | **8B ↬ 70B** | 90.00 | 70.00 | **92.00** | **84.00** | 73.33 | **81.33** | 66.67 | **78.67** | 75.00 | 77.76 |
| | **4B-width ↬ 70B** | 90.00 | 70.00 | 88.00 | 82.67 | **73.33** | 76.00 | **80.00** | 72.00 | **75.33** | **79.19** |
| | **4B-depth ↬ 70B** | 90.00 | 69.00 | 86.00 | 81.67 | 70.67 | 73.33 | 68.00 | 72.00 | 71.00 | 77.90 |
| wild west | 70B-zeroshot | 88.00 | **72.00** | 92.00 | **84.00** | 72.00 | 64.00 | 25.33 | 70.67 | 58.00 | 69.14 |
| | 70B-post-train | **91.00** | 68.00 | 90.00 | 83.00 | 69.33 | 65.33 | 61.33 | 68.00 | 66.00 | 73.29 |
| | **8B ↬ 70B** | 90.00 | 70.00 | **92.00** | **84.00** | **73.33** | **81.33** | 66.67 | **78.67** | 75.00 | **78.86** |
| | **4B-width ↬ 70B** | 90.00 | 70.00 | 88.00 | 82.67 | **73.33** | 76.00 | **80.00** | 72.00 | **75.33** | 78.48 |
| | **4B-depth ↬ 70B** | 90.00 | 69.00 | 86.00 | 81.67 | 70.67 | 73.33 | 68.00 | 72.00 | 71.00 | 75.57 |
| medieval castle | 70B-zeroshot | 88.00 | **71.00** | 89.00 | 82.67 | 62.67 | 74.67 | 20.00 | 73.33 | 57.67 | 68.38 |
| | 70B-post-train | 85.00 | 69.00 | 89.00 | 81.00 | 65.33 | 69.33 | 57.33 | 68.00 | 65.00 | 71.86 |
| | **8B ↬ 70B** | 90.00 | 72.00 | 89.00 | 83.67 | 72.00 | 76.00 | 68.00 | **84.00** | 75.00 | **78.71** |
| | **4B-width ↬ 70B** | **92.00** | 71.00 | **91.00** | **84.67** | **77.33** | **77.33** | **69.33** | 68.00 | **73.00** | 78.00 |
| | **4B-depth ↬ 70B** | 90.00 | 70.00 | 90.00 | 83.33 | 58.67 | 72.00 | 53.33 | 72.00 | 64.00 | 72.29 |

```
Andersen_fairy_tales_mappings = {        ancient_Egyptian_mappings = {
    "apartment": "cottage",                  "apartment": "palace",
    "bedroom": "chamber",                    "bedroom": "sleeping chamber",
    "bathroom": "washroom",                  "bathroom": "bathing room",
    "living room": "great hall",             "living room": "audience hall",
    "kitchen": "hearth",                     "kitchen": "kitchen",
    "coffeetable": "wooden table",           "coffeetable": "stone table",
    "desk": "writing desk",                  "desk": "writing table",
    "kitchentable": "feasting table",        "kitchentable": "dining table",
    "sofa": "wooden bench",                  "sofa": "cushioned bench",
    "kitchencabinet": "pantry",              "kitchencabinet": "storage chest",
    "cabinet": "cupboard",                   "cabinet": "treasure chest",
    "bathroomcabinet": "washstand",          "bathroomcabinet": "washstand",
    "dishwasher": "washing basin",           "dishwasher": "servant",
    "fridge": "cooling box",                 "fridge": "cool room",
    "microwave": "heating stone",            "microwave": "heating pot",
    "stove": "fireplace",                    "stove": "fire pit",
    "apple": "apple",                        "apple": "fruit",
    "book": "tome",                          "book": "papyrus scroll",
    "chips": "dried berries",                "chips": "flatbread",
    "condimentbottle": "spice jar",          "condimentbottle": "spice jar",
    "cupcake": "honey cake",                 "cupcake": "honey pastry",
    "dishbowl": "clay bowl",                 "dishbowl": "clay bowl",
    "plate": "wooden plate",                 "plate": "ceramic plate",
    "remotecontrol": "magic wand",           "remotecontrol": "scepter",
    "salmon": "smoked fish",                 "salmon": "dried fish",
    "waterglass": "goblet",                  "waterglass": "chalice",
    "wine": "mead",                          "wine": "wine",
    "wineglass": "goblet",                   "wineglass": "goblet"}
    "kitchencabinet": "pantry shelf"}
```

```
wild_west_mappings = {                   outer_space_mappings = {                 medieval_castle_mappings = {
    "apartment": "saloon",                   "apartment": "quarters",                 "apartment": "saloon",
    "bedroom": "bunk room",                  "bedroom": "sleeping quarters",          "bedroom": "bunk room",
    "bathroom": "outhouse",                  "bathroom": "sanitation room",           "bathroom": "outhouse",
    "living room": "bar area",               "living room": "recreation area",        "living room": "bar area",
    "kitchen": "cooking area",               "kitchen": "replicator station",         "kitchen": "cooking area",
    "coffeetable": "wooden table",           "coffeetable": "control console",        "coffeetable": "wooden table",
    "desk": "writing desk",                  "desk": "command station",               "desk": "writing desk",
    "kitchentable": "dining table",          "kitchentable": "mess table",            "kitchentable": "dining table",
    "sofa": "wooden bench",                  "sofa": "lounger",                       "sofa": "wooden bench",
    "kitchencabinet": "storage shelf",       "kitchencabinet": "storage unit",        "kitchencabinet": "storage shelf",
    "cabinet": "supply cabinet",             "cabinet": "storage unit",               "cabinet": "supply cabinet",
    "bathroomcabinet": "washstand",          "bathroomcabinet": "hygiene compartment","bathroomcabinet": "washstand",
    "dishwasher": "wash basin",              "dishwasher": "sterilizer unit",         "dishwasher": "wash basin",
    "fridge": "icebox",                      "fridge": "cold storage",                "fridge": "icebox",
    "microwave": "stove",                    "microwave": "food synthesizer",         "microwave": "stove",
    "stove": "wood stove",                   "stove": "heating unit",                 "stove": "wood stove",
    "apple": "fresh apple",                  "apple": "synthesized apple",            "apple": "fresh apple",
    "book": "ledger",                        "book": "data pad",                      "book": "ledger",
    "chips": "corn chips",                   "chips": "nutrition chips",              "chips": "corn chips",
    "condimentbottle": "sauce bottle",       "condimentbottle": "flavor vial",        "condimentbottle": "sauce bottle",
    "cupcake": "pastry",                     "cupcake": "synthesized pastry",         "cupcake": "pastry",
    "dishbowl": "ceramic bowl",              "dishbowl": "serving bowl",              "dishbowl": "ceramic bowl",
    "plate": "ceramic plate",                "plate": "serving plate",                "plate": "ceramic plate",
    "remotecontrol": "telegraph key",        "remotecontrol": "control pad",          "remotecontrol": "telegraph key",
    "salmon": "salted fish",                 "salmon": "replicated fish",             "salmon": "salted fish",
    "waterglass": "glass",                   "waterglass": "hydration vessel",        "waterglass": "glass",
    "wine": "whiskey",                       "wine": "synthesized wine",              "wine": "whiskey",
    "wineglass": "shot glass"}               "wineglass": "drinking vessel",          "wineglass": "shot glass"}
                                             "kitchencabinet": "storage unit"}
```

*Figure 7.* Primary changes from the VirtualHome simulator between the original apartment scenario and the five transferred thematic environments used in our ToM experiments.

*Table 12.* Performance on social interaction MuMA-ToM benchmark.

| Method | Belief | Social Goal | Belief of Goal | All |
|---|---|---|---|---|
| **Human** | 98.9 | 94.4 | 87.1 | 93.5 |
| Gemini 1.5 Flash | 53.9 | 33.0 | 41.4 | 42.7 |
| Gemini 1.5 Pro | 78.9 | 43.9 | 46.9 | 56.4 |
| Llava 1.6 13B | 70.2 | 43.2 | 17.9 | 43.7 |
| Llava 1.6 34B | **93.6** | 37.2 | 27.5 | 52.8 |
| GPT-4o | 67.9 | 39.6 | 44.4 | 50.6 |
| InternVL 2 26B | 59.3 | 44.9 | 35.5 | 46.6 |
| VideoLlama 2 7B | 70.1 | 45.6 | 37.7 | 51.1 |
| BIPALM llama2-7B | 41.2 | 34.1 | 30.6 | 33.9 |
| **LIMP (GPT-4o)** | **93.4** | **67.7** | **68.7** | **76.6** |
| **Ours (8B+405B)** | **94.0** | **64.5** | **67.5** | **75.3** |

*Table 13.* Performance comparison: Our weak-to-strong method vs. fully fine-tuned and PEFT baselines.

| Model | Fine-tuning Method | Data Size | LM Size | Acc. (%) |
|---|---|---|---|---|
| | FFT | 20,000 | 4B | 77.00 |
| Llama3.1-Minitron-4B-Width | LoRA | 20,000 | 4B | 76.90 |
| | 4bit-QLoRA | 20,000 | 4B | 76.33 |
| **Weak-to-Strong Control:** | | | | |
| | FFT-trained 4B | 20k | 70B | 78.67 |
| 4B-Width → Llama3.1-70B | LoRA-trained 4B | 20k | 70B | 78.52 |
| | 4bit-QLoRA-trained 4B | 20k | 70B | 78.10 |
| **PEFT on Large LM:** | | | | |
| | FFT | 20,000 | 70B | 71.45 |
| Llama3.1-70B | LoRA | 20,000 | 70B | 71.86 |
| | 4bit-QLoRA | 20,000 | 70B | 75.66 |

*Table 14.* Ablation study on the contribution of Weak-to-Strong (W2S) Control (with 8B as weak small LM).

| LM | Config | Belief Avg. | Goal Avg. | All Avg. |
|---|---|---|---|---|
| Llama2 70B | Post-trained (No W2S) | 82.33 | 72.00 | 76.43 |
| **Llama2 70B** | **Weak-to-Strong** | **83.00** | **74.33** | **78.05** |
| Llama3 70B | Post-trained (No W2S) | 83.33 | 65.33 | 73.05 |
| **Llama3 70B** | **Weak-to-Strong** | **86.00** | **73.33** | **78.76** |
| Llama3.1 70B | Post-trained (No W2S) | 85.00 | 62.00 | 71.86 |
| **Llama3.1 70B** | **Weak-to-Strong** | **85.67** | **74.67** | **79.38** |
| **Llama3.1 405B** | **Weak-to-Strong** | **87.10** | **77.14** | **81.29** |

